# Introducing Radiotherapy in Metastatic Merkel Cell Carcinoma Patients with Limited Progression on Avelumab: An Effective Step against Primary and Secondary Immune Resistance?

**DOI:** 10.3390/jpm13050841

**Published:** 2023-05-17

**Authors:** Gianluca Ferini, Valentina Zagardo, Paola Critelli, Anna Santacaterina, Serena Sava, Mandara Muralidhar Harikar, Tejas Venkataram, Giuseppe Emmanuele Umana, Anna Viola, Vito Valenti, Stefano Forte

**Affiliations:** 1REM Radioterapia srl, Via Penninazzo 11, 95029 Viagrande, Italy; valentina.zagardo@gmail.com (V.Z.); vito.valenti@grupposamed.com (V.V.); 2Department of Biomedical, Dental Science and Morphological and Functional Images, University of Messina, 98122 Messina, Italy; paola.critelli@unime.it; 3Radiation Oncology Unit, Papardo Teaching Hospital, 98158 Messina, Italy; anna.santacaterina@virgilio.it; 4Istituto Oncologico del Mediterraneo, 95029 Viagrande, Italy; serena.sava@grupposamed.com (S.S.); anna.viola@fondazioneiom.it (A.V.); stefano.forte@grupposamed.com (S.F.); 5Department of Neurosurgery, Trauma Center, Gamma Knife Center, Cannizzaro Hospital, 95126 Catania, Italy; mandara.harikar@gmail.com (M.M.H.); tejas.venkataram@gmail.com (T.V.); umana.nch@gmail.com (G.E.U.)

**Keywords:** Merkel cell carcinoma, oligometastasis, oligoprogressive, radiotherapy, stereotactic radiotherapy, immunotherapy, immune resistance

## Abstract

Purpose: To investigate the ability of radiotherapy (RT) to prolong progression-free survival (PFS) and to report treatment-related toxicities among oligoprogressive metastatic Merkel cell carcinoma (mMCC) patients on avelumab. Methods: We retrospectively collected clinical data on mMCC patients who underwent radiotherapy for limited progression on avelumab. Patients were categorized as primary or secondary immune refractory depending on the time of onset of resistance to immunotherapy (at the first or subsequent follow-up visits after avelumab initiation). Pre- and post-RT PFS were calculated. Overall survival (OS) from the first progression treated with RT was also reported. Radiological responses and toxicities were evaluated according to the irRECIST criteria and RTOG scoring system, respectively. Results: Eight patients, including five females, with a median age of 75 years, met our inclusion criteria. The median gross tumor and clinical target volumes at first progression on avelumab were 29.85 cc and 236.7 cc, respectively. The treatment sites included lymph node, skin, brain, and spine metastases. Four patients received more than one course of RT. Most patients were treated with palliative radiation doses (mainly 30 Gy in 3 Gy/day fractions). Two patients were treated with stereotactic RT. Five/eight patients were primary immune refractory. The objective response rate at the first post-RT assessment was 75%, whereas no local failure was reported. The median pre-RT PFS was 3 months. The pre-RT PFS was 37.5% at 6 months and 12.5% at 1 year. The median post-RT PFS was not reached. The post-RT PFS was 60% at 6 months and 1 year. The post-RT OS was 85.7% at 1 year and 64.3% at 2 years. No relevant treatment-related toxicity was observed. After a median follow-up of 18.5 months, 6/8 patients are still alive and continuing on avelumab therapy. Conclusions: Adding radiotherapy to mMCC patients with limited progression on avelumab seems to be safe and effective in prolonging the successful use of immunotherapy, regardless of the type of immune refractoriness.

## 1. Introduction

Merkel cell carcinoma (MCC) is a rare aggressive neuroendocrine skin tumor with a marked propensity for metastatic progression [1]. This occurrence translates into a dismal prognosis reaching a 1-year survival rate of about 38% [2]. The recent advent of immunotherapy, especially avelumab, has improved the prognosis of metastatic MCC (mMCC) [3]. In this stage, radiotherapy (RT) was mainly used for symptom palliation of painful metastases (either affecting the bones or not) or skin-ulcerating nodules or bulky mediastinal masses stenosing bronchi or great vessels [4]. In the JAVELIN trial, the avelumab arm reported 1-year progression-free and overall survival (OS) rates of 30% and 52%, which are significantly better than historical controls, although still not satisfactory [5]. Unlike platinum-based chemotherapy, avelumab produces more sustained responses after the first year of treatment [6]. The prolongation of PFS with immunotherapy raises some questions on the correct management of MCC patients with limited progressive disease, to whom the concepts of oligoprogression or oligorecurrence may be attributed. The latter refer to an intermediate disease extent between locoregional and polymetastatic ones, comprising up to five metastases, and is relatively novel for MCC, given its typical rapid progression in the previous chemotherapy era [7]. RT has proven to be effective in prolonging PFS in oligometastic tumors other than MCC [8]. Regarding this skin cancer, no reports are still available. In MCC patients with limited disease progression, RT might have a crucial role in reversing incoming refractoriness to immunotherapy, particularly in such a setting devoid of further effective systemic treatments. This use of RT has also been cautiously endorsed by a recent Delphi consensus, although no clinical validation exists [9].

Herein we describe a small series of patients with mMCC who had oligo-progression or -recurrence during avelumab treatment and then were treated with RT. We also provide a comprehensive overview of the literature on the subject.

## 2. Methods

We queried our mono-institutional database for mMCC patients undergoing RT from 2017 (approval date of avelumab for MCC in Italy) to 28 February 2023. Inclusion criteria were progression on avelumab and RT to all progressive disease sites, regardless of their extent and the RT technique used. Considering the peculiar biological behavior of the MCC, we did not screen the patients according to the concept of oligoprogression as commonly intended. Consequently, no restrictions on the number and size of irradiated metastases were applied. All fractionation schemes and RT techniques were welcome. Patients could have received cytotoxic chemotherapy prior to avelumab treatment or switched to it upon the completion of RT after immunotherapy failure. Exclusion criteria were radiotherapy delivered as adjuvant treatment for locoregional disease extent (stages I-III) or without at least one response assessment after avelumab initiation. No other immune checkpoint inhibitors were possible since avelumab is the only one approved for MCC in our country.

Regarding the RT target delineation, the gross tumor volume (GTV) was the progressive macroscopic disease as detected on the computed tomography (CT) simulation with the support of 18F-FDG positron emission tomography (PET), if any. GTV was included in a larger volume accounting for the probable subclinical spread of tumor cells, namely the clinical target volume (CTV), e.g., the entire inguinal lymph node basin in case of an isolated metastasis located therein. Given the lack of precise instructions to delineate the CTV within the context of the study, a certain arbitrariness was allowed. In cases of stereotactic radiosurgery (SRS) for brain metastases, only the GTV and planning target volume (PTV) were contoured. When stereotactic body radiation therapy (SBRT) was used for extracranial targets, the CTV delineation was at the discretion of the treating physician, unless otherwise stated by any consensus papers [10].

Patients were categorized as primary or secondary immune-refractory depending on the time of onset of resistance to avelumab: progressive disease (PD) at the first follow-up visit after avelumab initiation (requiring per-protocol confirmation as specified below) vs. PD at any other follow-up. These visits were scheduled every three months and included both physical and instrumental exams (CT and/or PET).

A series of demographic and treatment data was collected, including ECOG performance status, baseline immune status (immunocompetent vs. immunosuppressed), total radiation dose and fractionation, biologically effective dose (BED) using an α/β value of 10 Gy for tumor, number of RT courses, RT treatment site, GTV and CTV size, number of treated metastases, RT technique, further systemic treatments (before or after avelumab therapy), immune sensitivity at the time of RT (primary vs. secondary refractory), survival state at the date of the last follow-up (alive or not on 28 February 2023). The presence of the Merkel cell polyomavirus in the tumor genome as well as the PD-L1 status were not routinely investigated according to our standard clinical practice and should be considered unknown for all patients.

The study endpoints were the post-RT assessment of local control (LC), progression-free survival (at any site), overall survival, and the objective response rate (ORR). Radiological responses were defined as complete response (CR), partial response (PR), stable disease (SD), and progressive disease (PD), and assessed as per irRECIST criteria, according to which any first progression must be confirmed by repeating the instrumental evaluation at least four weeks later [11]. We also reported pre-RT PFS, which is the time interval between avelumab initiation and subsequent RT-treated progression. Toxicities were graded according to RTOG [12].

Univariate probabilities of post-RT OS and PFS were calculated using the Kaplan–Meier method [13]. The analyses were performed using R V.4.1.1 (R package survival).

## 3. Results

Our final cohort comprised eight patients (three males, five females) whose age ranged from 61 to 86 (median 75, interquartile range 16.25). No patients were immunosuppressed as none had immunodeficiency disease or received immunosuppressive drugs for any autoimmune conditions. The median total radiation dose was 30 Gy (range 19.5–36) in 10 fractions (range 3–12), corresponding to a median BED of 39 Gy (range 32.2–59.5). The number of RT courses was one for four patients, two in the other three, and four in one patient. The treatment sites included lymph node, skin, brain and spine metastases. The median GTV and CTV sizes at the first RT course were 29.85 cc (range 10–209) and 236.7 cc (range 37–516), respectively, encompassing a median of one metastasis (range 1–4). Overall, GTV and CTV ranged from 10 to 209 cc (median 29.35 cc) and from 25 to 3744 cc (median 236.7 cc), respectively, while the number of metastases treated simultaneously was 15 in the fourth course of one patient. Volumetric modulated arc therapy was used in six patients, while SRS and SBRT were used in the other two. Avelumab was first-line treatment in all but two patients who had previously been treated with cisplatin-based chemotherapy. Five/eight patients were to be considered as primary immune refractory. Six/eight were alive at the last follow-up. The above information is summarized in Table 1.

ORR (PR + CR) at the first post-RT assessment was 75%. Only two patients had SD. No patient exhibited in-field failure (LC equal to 100%).

The highest acute toxicity was a self-limiting G1 skin erythema in one patient. No treatment-related chronic toxicities were reported.

Starting from the date of the first RT, the median follow-up was 18.5 months (range 4–63). The median pre-RT PFS was 3 months as a result of having more than half of the patients with primary immune resistance. The pre-RT PFS was 37.5% at 6 months and 12.5% at 1 year. The median post-RT PFS was not reached as only four patients experienced further progression after RT administration. The post-RT PFS was 60% at 6 months and 1 year. The post-RT OS was 85.7% at 1 year and 64.3% at 2 years. The survival curves are in Figure 1, Figure 2 and Figure 3. The disease course of each patient is graphically displayed in Figure 4. Given the small sample size, no CTV or GTV thresholds can be detected to predict any association with survival outcomes.

Avelumab therapy was successfully maintained and is still ongoing in all but two patients who briefly switched to cytotoxic chemotherapy at the third and fifth progression, respectively, ultimately dying of the disease.

## 4. Discussion

To our knowledge, this is the first short report describing the integration of radiotherapy in the setting of MCC patients progressing during avelumab treatment. We reported promising results in terms of LC and PFS and no significant toxicities. To date, oligometastatic status for MCC has been only conceptualized, being still clinically unproven. Indeed, in the SABR-COMET trials, this condition has been described for the most common solid malignancies including, above all, breast, prostate, colorectal and lung cancers [8]. A recent trial is expanding this concept even to other tumors [14]; among these, the only skin-derived cancer was melanoma, already included in other similarly conceived studies [15,16]. Case reports hypothesized the oligometastatic stage also for cutaneous squamous cell carcinoma [17], which has mostly local aggressiveness like the basal cell one [18,19].

While oligometastatic disease generally refers to a condition starting with few metastases (≤5) diagnosed synchronously to the primary tumor uncontrolled, oligoprogression and oligorecurrence are characterized by a low tumor burden refractory to ongoing therapies (metachronous metastases). These disease stages are still amenable to metastases-directed local therapies, which allow the need for switching the ongoing systemic treatment to be postponed [20]. This is particularly important in the scenario investigated here, since the PD-1/PD-L1 blockade represents the last line of treatment, given the limited and transient effectiveness of classic cytotoxic chemotherapies. In our case series, 6/8 patients continued an effective use of avelumab after radiotherapy.

Oligometastatic status is generally linked to the use of stereotactic body radiation therapy (SBRT) to control with ablative radiation doses in a few daily fractions (≤5) no more than five metastases, usually not exceeding 5 cm in diameter [21]. We have chosen not to strictly adhere to this rigid definition for several reasons: (1) MCC is a greatly radiosensitive tumor responding also to radiation doses much lower than those used in SBRT; (2) MCC may progress along a wide skin area developing more than five nodules or a lymph node basin producing bulky masses greater than 5 cm, but still amenable to radiotherapy; (3) MCC has a typical subclinical spread between contiguous metastases, which often justifies the irradiation of a CTV much larger than the GTV. These facts make the classic dimensional and numerical criteria inadequate in defining the oligometastatic status for MCC, which likely needs ad hoc considerations. Our GTVs ranged from 10 to 209 cc, while CTVs from 25 to 3744 cc. The largest CTV refers to the whole skin of the left lower limb of a patient in whom previous three RT courses had failed to prolong PFS, as described in [22]. Palliative doses were the most common, having been used in 6/8 patients: by delivering a 3 Gy/day fraction, a total dose of 30 Gy was administered to 5 patients and 36 Gy to another one. Among these, local tumor regression was observed in all patients. The decision to subject two patients to SBRT rather than to the classic palliative radiotherapy treatments depended on the location (one brain, one vertebra, and two skin nodules) and on the small size of the metastases. Intriguingly, one of these two patients exhibited no objective response at the second course. Overall, four patients had further progressions treated with RT after the first course: in two patients, disease control was effectively restored, whereas, in the other two, RT was unable to stop a rapidly evolving systemic progression, ultimately culminating in death despite a last-ditch attempt to switch from avelumab to cisplatin-based chemotherapy (Figure 4).

Concerning the efficacy of low-dose palliative radiation, our results agree with those of Iyer et al. [23], who reported a durable objective response in 94% of patients with mMCC (including 45% of CR) treated with 8 Gy-single-fraction RT. Even these authors claimed a convenient synergy between RT and the host immune system (when not suppressed due to comorbidities). Both low (2–4 Gy per fraction) and high (>5 Gy per fraction) radiation doses may work as immune stimulants, the former by upregulating the major histocompatibility complex, the latter by increasing antigen release and priming T cells into the tumor (as well as directly inducing cell death by apoptosis or necrosis) [24,25,26]. Furthermore, very low doses (0.5–1 Gy per fraction) counteract the tumor immune desertification by mobilizing T cells from draining lymph nodes, thus enhancing tumor responsiveness to combined immunotherapy (IT) [25]. Irradiating a CTV significantly larger than GTV, as in most of our cases, could have a dual function: 1) to sterilize any microscopic tumor deposits in the area surrounding the GTV; 2) to extend the off-target volume receiving the immunoactivating low doses. The latter antitumor effect might be weakened by radiation-induced lymphopenia, thus requiring caution in delineating CTV and using dose-scattering intensity-modulated radiotherapy (IMRT) techniques [27]. The small size of our sample prevented us from providing recommendations on CTV delineation. According to our report, classic palliative RT schemes appear sufficient and adequate to achieve durable tumor control, making the use of time-consuming and costly SBRT unnecessary. It should be noted that none of our patients was treated with 8 Gy single-fraction RT: although Iyer et al. reported excellent results with this dosage, other authors doubt its adequacy and suggest tripling its administration to obtain better outcomes [23,28]. In our clinical practice, when the patient’s life expectancy is longer than six months, multi-fraction palliative schemes are preferred over the 8 Gy single fraction as the latter carries a higher risk of need for re-treatment [29].

Most of our patients had limited disease progression (median GTV < 30 cc) on IT. The mechanism underlying this acquired immune resistance probably involves localized defects in tumor antigen presentation and recognition by the host immune system [30]. RT may ablate the refractory focus, prolonging tumor response to IT [31]. This happened in most of our patients. Our preliminary results introduce MCC among oligoprogressive cancer histologies benefitting from RT to reverse incoming immunologically cold tumor subpopulations [32,33].

A series of interesting findings emerges from reviewing the pertinent literature. In cases of limited brain metastases, SRS, as characterized by less cognitive deteriorating effects than whole-brain RT, might be preferred over this [34]. In the pre-avelumab era, Jacob et al. reported a successful use of SRS for two metachronous brain metastases in an MCC patient who subsequently died of leptomeningeal spread [35]. Stereotactic RT showed excellent local control in another patient with limited brain metastases from a primary parotid Merkel-type small cell neuroendocrine carcinoma, which is indistinguishable from classic MCC [36]. In the latter case, avelumab would have been initiated as a salvage treatment, given the failure of chemotherapy and the limited efficacy of RT in significantly prolonging the patient’s PFS. Regarding intracranial oligoprogression during avelumab treatment, a brief report by Fife et al. promotes the addition of SRS to IT: avelumab’s effects on intracranial metastases were transient or even null in all four patients, while two of them actually benefitted from brain SRS [37]. The outcome of our single case of intracranial metastasis, together with all the above, further supports the use of brain SRS.

Concerning the management of extracranial oligo-progression or -recurrence from MCC, there are few reports with both SBRT and non-stereotactic schemes. SBRT was effectively used at a dose of 25 Gy (with a hotspot of 33.57 Gy) in 5 fractions every other day for a challenging intracardiac MCC metastasis, which arose after a durable complete response to avelumab, lasting 2 years and 5 months. The subsequent treatment success allowed safely resuming of IT, which ensured excellent systemic disease control until the last documented follow-up (almost 1 year after SBRT delivery) [38]. A similar course was observed in an HIV-positive patient, who, while switching to avelumab, received SBRT at a dose of 30 Gy in six daily fractions to a para-cardiac lymph node metastasis progressing on pembrolizumab therapy and was still progression-free 8 months later [39]. Large doses in a few fractions can also be particularly useful in some extremely palliative settings that require rapid and sustained relief of symptoms and a reduction in daily visits to the oncological center, such as in medically frail or elderly patients [18,40]. On the other side, conventionally fractionated radiotherapy was able to restore sensitivity to IT in two MCC patients, whose metastases had become late refractory to pembrolizumab and avelumab, respectively. At the last follow-up after RT, both patients were alive, achieving an exceptionally long PFS (20 months and 2.5 years, respectively) [41,42].

RT is known to synergize with the antitumor effects of PD1/PD-L1 axis blockade [43,44]. This probably underlies some tumor responses that extend beyond the radiation field, as in the case series by Xu et al. These authors reported the impressive outcomes of two patients with mMCC progressing on PD-1 inhibitor therapy and then treated with 8 Gy-single-fraction palliative radiotherapy to some portion of the tumor masses: surprisingly, both patients rapidly experienced a marked reduction until disappearance not only of the irradiated tumor portion, but also of the distant out-of-field tumor sites [45]. In fact, the abscopal effect has been rarely reported, and no way is yet available to systematically engage it to enhance systemic tumor control, being able to be elicited even in the absence of IT [46]. In the NCT03071406 trial, SBRT delivery to a tumor site did not produce a significant response in non-irradiated ones among mMCC patients treated with ipilimumab plus nivolumab, ultimately resulting in no ORR difference compared to IT alone [47]. This is why we decided to focus on those patients for whom it was possible to irradiate all macroscopic disease sites.

On the other hand, i.e., from the point of view of toxicity, it is well-known that combining RT with IT, as well as with other drugs, may give rise to unexpected events, such as the bystander effect, radiation recall phenomena, or even life-threatening autoimmune paraneoplastic disorders [48,49,50]. In our small cohort, we did not report any significant higher-than-expected toxicity.

A very interesting finding of our investigation concerns the proven ability of RT to reverse both primary and secondary immune resistance. Indeed, three out of five patients with primary immune resistance showed sustained PFS following RT to the immune-refractory tumor sites detected at the first follow-up after avelumab initiation. Furthermore, RT was able to effectively prolong PFS even in those three patients who had secondarily developed immune resistance to avelumab. Notably, RT was delivered a second time in two of these patients, allowing them to successfully continue avelumab therapy. These results are of outstanding importance if we consider that a short report from the ADOREG registry showed an ORR of 50% (7/14) in avelumab-refractory mMCC patients later treated with ipilimumab plus nivolumab and immune-related toxicities requiring treatment discontinuation in 21% (3/14) of them [51]. Our findings mean that the possible positive interactions between RT and IT in mMCC patients deserve further attention from clinicians and researchers.

Successful interactions between RT and IT have been recently reviewed by Yi et al., but their summary does not include MCC histology [52]. Furthermore, some preclinical models could clarify whether personalization of stereotactic radiotherapy in combination with IT is needed [53]. Several lines of evidence advocate the use of RT for MCC, but none in the context explored here [54,55].

The main limitations of our study are the small number of patients and the heterogeneity of the target size and dose fractionation, which did not make it possible to draw clear conclusions about the most suitable RT approach. However, based on the favorable benefit–risk ratio, adding radiotherapy for rescuing mMCC patients with limited progression on avelumab might be a compelling strategy. The encouraging findings reported here should not be ignored and warrant further investigation in large prospective trials, possibly of the multicenter type to overcome the patient recruitment difficulties linked to the tumor rarity.

MCC is rare and the oligoprogression brought to the attention of radiation oncologists is even more so. Our retrospective observations from a single institution would translate into a prospective recruitment rate of approximately 1.33 patients per year. The latter, considering avelumab’s well-proven efficacy and its recent approval as factors limiting the development of the disease stage investigated here, may be in line with the all-stage rate reported by Wang et al. over forty years, equal to about 13 patients per year [56]. Spreading awareness of our preliminary results among medical oncologists could lead to an increase in our low recruitment rate. Meanwhile, our findings aim to serve as a hypothesis-generating and -testing proof-of-concept study.

## 5. Conclusions

Radiotherapy seems to be well tolerated and effective in prolonging progression-free survival in mMCC patients with limited progression on avelumab, thus allowing its continuation. In achieving that goal, low radiation doses, such as those used in palliative schemes, might be as equally effective as high ablative ones, with even fewer concerns about treatment tolerance. However, the suggestive data presented here need to be validated in large prospective trials, pending new drugs that will hopefully extend the therapeutic algorithm for the metastatic stage of such aggressive cancer.

## Figures and Tables

**Figure 1 jpm-13-00841-f001:**
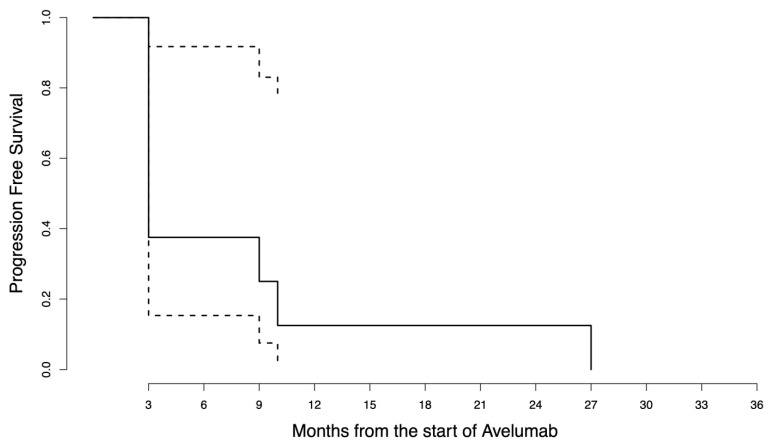
Pre-radiotherapy progression-free survival curve. The survival function is represented by the solid line, where the steps represent the events that have occurred (first disease progression after avelumab initiation). The dashed lines represent the 95% confidence interval.

**Figure 2 jpm-13-00841-f002:**
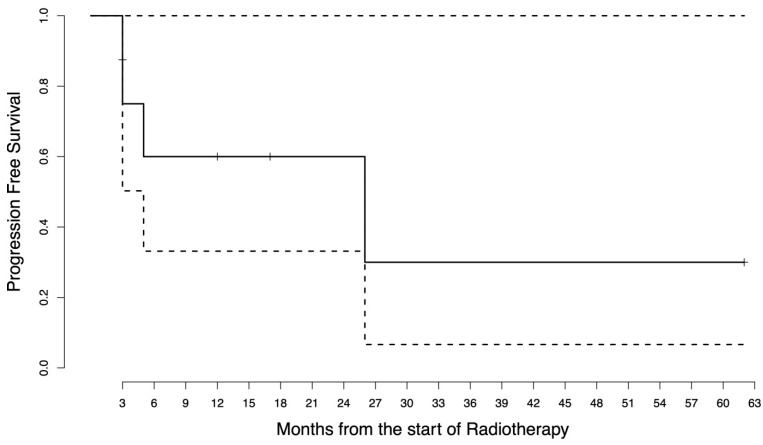
Post-radiotherapy progression-free survival curve. The survival function is represented by the solid line, where the steps represent the events that have occurred (first disease progression after radiotherapy) while the vertical bars represent the censored patients. The dashed lines represent the 95% confidence interval.

**Figure 3 jpm-13-00841-f003:**
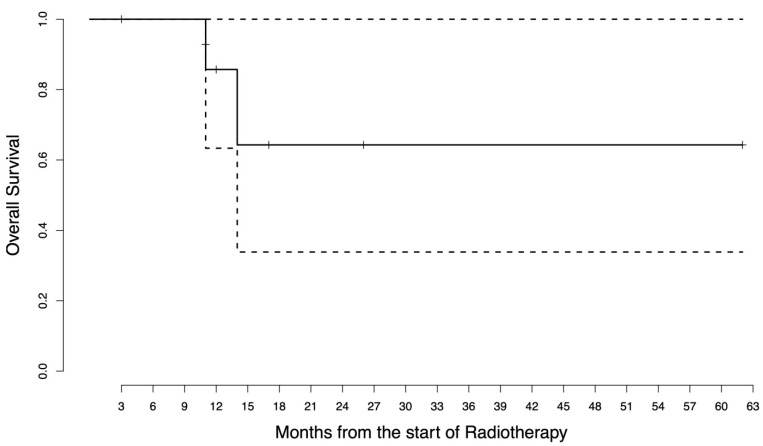
Post-radiotherapy overall survival curve. The survival function is represented by the solid line, where the steps represent the events that have occurred (death) while the vertical bars represent the censored patients. The dashed lines represent the 95% confidence interval.

**Figure 4 jpm-13-00841-f004:**
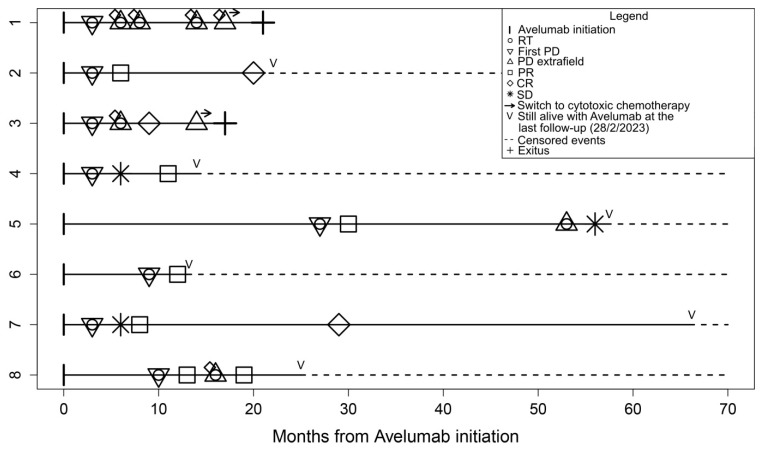
Timeline of all patients included with the explanation of the events in the legend. RT, radiotherapy; PD, progressive disease; PR, partial response; CR, complete response; SD, stable disease. For patients 5 and 8, two cycles of SBRT each were used; the others were treated with palliative RT schemes (the circles stand for RT courses, as explained in the figure legend).

**Table 1 jpm-13-00841-t001:** Patients’ characteristics. RT, radiotherapy; GTV, gross tumor volume; CTV, clinical target volume; VMAT, volumetric modulated arc therapy; SRS, stereotactic radiosurgery; SBRT, stereotactic body radiation therapy.

	Patients % (*n*)
Age, median (range) years	75 (61–86)
Sex	
Male	37.5 (3/8)
Female	62.5 (5/8)
Median Total Radiation Dose	30 (19.5–36)
Number of RT courses	
1	50 (4/8)
2	37.5 (3/8)
4	12.5 (1/8)
Type of treatment	
VMAT	75 (6/8)
SRS and SBRT	25 (2/8)
Treatment before Avelumab	
None	75 (6/8)
Cisplatin-based chemotherapy	25 (2/8)
Treatment after Avelumab	
None	75 (6/8)
Cisplatin-based chemotherapy	25 (2/8)
Primary immune refractory patients	62.5 (5/8)
Patients alive at last follow-up	75 (6/8)
Patients who had local control	100 (8/8)
	Median (range)
Median GTV at the first RT course, cc	29.85 (10–209)
Median CTV at the first RT course, cc	236.7 (37–516)
Median overall GTV, cc	29.35 (10–209)
Median overall CTV, cc	236.7 (25–3744)
Median number of simultaneously RT-treated metastases	1 (1–15)

## Data Availability

All the generated data are available upon request to the corresponding author.

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
