# Peer review of "Introducing Radiotherapy in Metastatic Merkel Cell Carcinoma Patients with Limited Progression on Avelumab: An Effective Step against Primary and Secondary Immune Resistance?"

_jpm, 2023, doi:10.3390/jpm13050841_

Round 1
Reviewer 1 Report
First, the n number is very small, 2 of them were treated with a different type of radiotherapy (stereotactic). Four received more than one course or RT, you mention MOST patients received palliative radiation, then why not all?. These are probably factors that were not controlled by the authors. However, these groups are not comparable, given RT was different in dose and type and frequency. Therefore, it is not clear to which groups corresponds the results observed. Is it difficult to attribute the results to radiotherapy given that all situations are different. Any conclusions made would not be valid.
Why were all progressive disease sites, regardless of their extent and the RT technique used included, whereas locoregional disease extent was not? These last ones are also resistant, aren’t they?
There are too many acronyms and that makes it very difficult to follow.
Line 30 – Please state the measure in Gy for palliative radiation.
Line 106 add the interquartile range.
Line 107 - How was immunosuppression measured? Was this extracted from the record? Or measured somehow?
In Table 1 Instead of putting 4/8, 3/8 etc., a % should be given.
In Table 1, for the median and ranges another line should be added above where the units that correspond to 28.85 (10-209) should written. (Median (range)) These could be moved to the end of the table so for all the other variables you still use %(n)
Table 1 - Add the meaning of the acronyms at the foot of the table. Tables should be able to stand alone, and the reader should be able to understand it by itself without going back to the text to find out the meaning of acronyms.
Table 1- Remove patients with treatment-related chronic toxicity if not relevant for your results.
Line 127 - I assume OS is overall survival, however this acronym is not described anywhere else.
- In Figure 4 Add the meaning of the acronyms (same for all figures).
- SD acronym is commonly used for standard deviation, I would suggest choosing another acronym.
Discussion:
Line 134 – there are other reports about radiotherapy in patients with chemotherapy and avelumab: https://pubmed.ncbi.nlm.nih.gov/36276497/
RT has also been suggested in an update in 2017 https://www.ncbi.nlm.nih.gov/pmc/articles/PMC5690804/
https://pubmed.ncbi.nlm.nih.gov/33794306/
https://pubmed.ncbi.nlm.nih.gov/35062949/
Line 136 – nihil.
Line 165 - Palliative doses are still not being described up to this point.
Line 166 – What is mMCC?
FIGURE 2 and 3 should be modified as not all the patients were followed-up for 63 months and this figure gives the impression that this was the case. Furthermore, that they have survived for 63 months or so, whilst only one patients was followed for 63 months. The median was 18.5 months of follow up. Another kind of figures should be used. Or this figures should be removed. It is clear from the text that 2 have died. Four patients were only followed for around 20 months.
Discussion needs English improvement and rewriting in some parts.
It seems discussion was centered on the definition of oligometastasis and the justification of the use of this term in the present project. However, the discussion should be centered in the use of radiotherapy In other cancers and MCC should be developed.
Rephrase the following sentences:
Line 139 - More recent trials are expanding this concept even to other tumors like the kidney one, but still ignoring 139 those originating from the skin [14]. As regards these latter, oligometastatic disease has been described for 140 melanoma [15, 16]. Case reports hypothesized this condition also for cutaneous squamous cell carcinoma 141 [17], which has mostly local aggressiveness like the basal cell one [18, 19]. ß Here, you mention tumors originating from skin are being ignored, however they haven’t been ignored in literature. These sentences could be merged. Something along the lines: “Recent trials are expanding this concept to other tumors such as kidney (14), melanoma (15, 16), and squamous cell carcinoma (17). The latter characterized by a local aggressiveness similar to the basal cell carcinoma[18, 19].”
Line 146 - which allow the postponing of the need for 146 switching systemic treatments [20]
Line 156 - but still amenable to involved field RT;
Author Response
Dear reviewer,
you find attached our reply to your comments.
Thank you for your insightful comments

Reviewer 2 Report
The problem of drug resistance in the treatment of oncological neoplasms concerns many localizations of this pathology. Metastatic Merkel cell 2 carcinoma refers to extremely aggressive neoplasms.
The use of radiotherapy in patients with mMCC treated with avelumab has been shown to be successful.
This study describes a small sample of patients with mMCC who relapsed during treatment with avelumab followed by RT.
Several questions arose
1. It should be specified in more detail why radiotherapy was used for the treatment of patients immunoresistant to avelumab with different courses from 1 to 4 radiations?
2. Could resistance have developed in patients who received cisplatin therapy before using avelumab?
3. In your opinion, which therapy scheme can be considered the most successful for this disease?
4. And is there a difference in the success rate of this combined therapy scheme by gender?
Also, wishes regarding the presented graphs: for a better perception of the drawings, it is worth increasing the font and the quality of the images.
Author Response
Dear reviewer,
you find attached our reply to your comments.
Thank you for your insightful comments.

Round 2
Reviewer 1 Report
With regards to the Figures 1, keep the meaning of ICI.
Figure 1 I think the legend should be Post-avelumab or ICI as you wish, instead of pre-radiotherapy, as the time pre-radiotherapy is not shown/known, your starting point is avelumab. In both figure 1 and 4.
Figure 1, 2, and 3. State what the dashed lines represent. State on your figures the meaning of vertical lines. An example can be found here: https://www.ncbi.nlm.nih.gov/pmc/articles/PMC3059453/
Figures 1, 2 and 3, you must at least put the end point of each patient. You mentioned the censored events are the vertical lines, however in figure 2, there are 4 vertical short lines and 2 deaths (=6, not matching the 8 patients). In this same figure, you have 3 long vertical lines, which are your deaths, however you have only reported 2 deaths.
Author Response
Reviewer's comment: With regards to the Figures 1, keep the meaning of ICI.
Authors' reply: Unfortunately, the editor assistant who we asked to upload the zipped figures in the manuscript didn't make it. We deleted ICI because we used Avelumab. ICI is useless since it wasn't used otherwhere. Now, we correctly uploaded the revised figures in the manuscript. Please, check them.
Reviewer's comment: Figure 1 I think the legend should be Post-avelumab or ICI as you wish, instead of pre-radiotherapy, as the time pre-radiotherapy is not shown/known, your starting point is avelumab. In both figure 1 and 4.
Authors' reply: we'd prefer to maintain the current nomenclature using "pre-radiotherapy" since the main topic around which the paper turns is radiotherapy. However, the concordance with the post-avelumab initiation interval is inferable by looking at the X-axis of figure 1, where we used Avelumab.
Reviewer's comment: Figure 1, 2, and 3. State what the dashed lines represent. State on your figures the meaning of vertical lines. An example can be found here: https://www.ncbi.nlm.nih.gov/pmc/articles/PMC3059453/
Authors' reply: We clearly specified what you required in the figures' captions. Please, check them.
Reviewer's comment: Figures 1, 2 and 3, you must at least put the end point of each patient. You mentioned the censored events are the vertical lines, however in figure 2, there are 4 vertical short lines and 2 deaths (=6, not matching the 8 patients). In this same figure, you have 3 long vertical lines, which are your deaths, however you have only reported 2 deaths.
Authors' reply: Dear reviewer, we checked several times the correctness of our graphs and there are no mistakes. Figure 2 represent a Kaplan Meier analysis of progression free survival. Long vertical lines thus represent events of progression, while short vertical ticks on the graph represent right censored patients. At month 3 there are 3 events: two progressions (which constitute the long vertical line that represent a decrease in the number of progression free patients) and 1 censor (because a patient is still progression free at the end of its 3 months follow-up). At month 5 a patient incurred in a progression event, so a vertical line represents a proportional decrease of progression free patients. At months 12 and 17 there are two more patients that were still progression free at the end of their follow-up (which were respectively 12 and 17 months) so there are two vertical ticks at the corresponding times. Then at month 26 other patients incurred in a progression, so we have another decrease in the proportion of progression free patients that is represented by the long vertical line. The extent of this line is related to the variation of the proportion between patients at risk (so patients which are progression free and still in follow-up at that time) and patients with the progression event at that time, according to the Kaplan-Meier method. After month 26 there are no other progression events, and the remaining patient remains progression free until the end of his follow-up at month 62 where a vertical tick indicates censor.